# Stress-Induced Transcriptomic Changes in Females with Myalgic Encephalomyelitis/Chronic Fatigue Syndrome Reveal Disrupted Immune Signatures

**DOI:** 10.3390/ijms24032698

**Published:** 2023-01-31

**Authors:** Derek J. Van Booven, Jackson Gamer, Andrew Joseph, Melanie Perez, Oskar Zarnowski, Meha Pandya, Fanny Collado, Nancy Klimas, Elisa Oltra, Lubov Nathanson

**Affiliations:** 1John P. Hussman Institute for Human Genomics, Miller School of Medicine, University of Miami, Miami, FL 33136, USA; 2Institute for Neuro-Immune Medicine, Dr. Kiran C. Patel College of Osteopathic Medicine, Nova Southeastern University, Fort Lauderdale, FL 33328, USA; 3Dr. Kiran C. Patel College of Osteopathic Medicine, Nova Southeastern University, Fort Lauderdale, FL 33328, USA; 4Halmos College of Arts and Sciences, Nova Southeastern University, Fort Lauderdale, FL 33328, USA; 5Farquhar Honors College, Nova Southeastern University, Fort Lauderdale, FL 33328, USA; 6Department of Veterans Affairs, Miami VA Healthcare System, Research Service, Miami, FL 33125, USA; 7South Florida Veterans Affairs Foundation for Research and Education Inc., Fort Lauderdale, FL 33125, USA; 8School of Medicine, Universidad Católica de Valencia San Vicente Mártir, 46001 Valencia, Spain

**Keywords:** myalgic encephalomyelitis, chronic fatigue syndrome, transcriptomics, dysregulated immune pathways, post-exertional malaise

## Abstract

Myalgic encephalomyelitis/chronic fatigue syndrome (ME/CFS) is a chronic, complex multi-organ illness characterized by unexplained debilitating fatigue and post-exertional malaise (PEM), which is defined as a worsening of symptoms following even minor physical or mental exertion. Our study aimed to evaluate transcriptomic changes in ME/CFS female patients undergoing an exercise challenge intended to precipitate PEM. Our time points (baseline before exercise challenge, the point of maximal exertion, and after an exercise challenge) allowed for the exploration of the transcriptomic response to exercise and recovery in female patients with ME/CFS, as compared to healthy controls (HCs). Under maximal exertion, ME/CFS patients did not show significant changes in gene expression, while HCs demonstrated altered functional gene networks related to signaling and integral functions of their immune cells. During the recovery period (commonly during onset of PEM), female ME/CFS patients showed dysregulated immune signaling pathways and dysfunctional cellular responses to stress. The unique functional pathways identified provide a foundation for future research efforts into the disease, as well as for potential targeted treatment options.

## 1. Introduction

Myalgic encephalomyelitis/chronic fatigue syndrome (ME/CFS) is a complex, chronic, multi-system illness characterized by fatigue, autonomic dysfunction, cardiovascular and gastrointestinal issues, neurological disturbances, and muscle/joint pain [1]. ME/CFS has been estimated to affect between 836,000 and 2.5 million Americans, with women being affected at a rate of three times that of men [2]. One key feature of ME/CFS is post-exertional malaise (PEM). PEM typically occurs after physical exertion or cognitive/emotional stimuli. It often involves a loss of physical or mental stamina and/or rapid muscle fatigability and can persist for days or even weeks [3]. These episodes can be debilitating, preventing patients from performing normal daily activities, with an estimated 25 percent being completely housebound or bedbound throughout both relapses and remissions [2].

Currently, there are no reliable biomarkers that are used to diagnose ME/CFS. Diagnosis is made by identifying patterns of symptoms utilizing criteria established by the US National Academy of Medicine in conjunction with criteria from other medical bodies [1]. Treatment focuses on symptom amelioration due to poor comprehension of the pathophysiology underlying this condition. As such, advancing our understanding of ME/CFS is critical for the development of diagnostic biomarkers and more targeted therapies. 

Over the past few decades, research efforts have documented underlying biological disturbances in patients with ME/CFS, including the dysfunction of the central nervous system and immune system, and the dysregulation of cellular energy metabolism [4,5,6]. More recently, advances in omics technology have enabled researchers to delve deeper into the underlying mechanisms potentially driving these abnormalities. For example, genomic studies, including GWAS, have found specific polymorphisms and SNPs [7,8] that may predispose individuals to the disease. The evidence of epigenetic alterations, including distinct DNA methylation patterns in immune cells and differentially expressed miRNAs, has been associated with ME/CFS [9,10,11,12,13,14]. Proteomic studies have elucidated subsets of immune proteins that differ between ME/CFS patients and controls in plasma and cerebrospinal fluid [15,16,17,18]. Studies involving the metabolomics of plasma, fecal bacteria, and peroxisomes have revealed irregularities in lipid, amino acid, and steroid metabolism among others [19,20,21,22]. Pioneering work in the field of transcriptomics has highlighted how dysregulated circadian rhythm pathways contribute to faulty cellular metabolism and how altered receptor signatures in lymphocytes play a role in immune system dysfunction [23,24].

These distinct changes in ME/CFS patients have been reported at rest and are not related to exertion. The most devastating effects of the disease, however, often present in response to stress. By subjecting patients to an exercise challenge, researchers can take a deeper look into the underlying biological processes associated with fatigue and PEM. Unique metabolomic pathways have been characterized in ME/CFS patients before and after exercise challenge [25]. In the field of transcriptomics, one study attempted to delineate gene expression changes in whole blood of ME/CFS patients in response to exercise [26]. In that study, no significant differences in gene expression were found between ME/CFS patients and healthy controls (HC), which may be attributed to the heterogeneity of whole blood components. To further characterize temporal transcriptomic changes in ME/CFS patients in response to an exercise challenge, our study focuses on immune cells isolated from whole blood cells called peripheral blood mononuclear cells (PBMCs). These cells are primarily made up of lymphocytes (B cells, T cells, and NK cells) and also include monocytes and dendritic cells [27].

Our research team utilized an exercise challenge meant to provoke PEM in female ME/CFS patients and then analyzed changes in gene expression using RNA sequencing (RNA-seq) at three different time points: baseline before exercise challenge (T0), maximal exertion (T1), and 4 h after maximal exertion (T2). The results from this study provide a better understanding of the pathophysiology behind ME/CFS females, with the goal of advancing diagnostic techniques and developing more targeted therapeutic options.

## 2. Results

### 2.1. Participant Characteristics

This study compared 20 female ME/CFS patients and 20 matched female HCs who underwent an exercise challenge. Participant characteristics were obtained at the time of blood draw before exercise challenge. There was no significant difference (*p* > 0.05) in age and BMI between individuals with ME/CFS and HCs (Table 1).

Physical health and mental health were evaluated based on the answers to the short form 36-item survey (SF-36) questionnaire [28]. The ME/CFS cohort had poor self-reported outcomes (*p* < 0.05) in all domains of SF-36 (except “Role Emotional”), as compares to the level of disability (Table 1). However, all participants were physically capable of completing the exercise challenge.

### 2.2. Transcriptomic Changes between Maximal Exertion (T1) and Baseline before Exercise Challenge (T0)

We analyzed gene expression in PBMCs in both ME/CFS patients and HCs between two time points, T1 (maximal exertion) and T0 (baseline before exercise challenge). In HCs, there were 102 genes that showed significant changes in expression: 4 genes were underexpressed (PHLDB3, LZTS3, SLC16A10, MAL) and 98 were overexpressed (Appendix A). 

The four most significant clusters of differentially expressed genes (DEGs) categorized by pathway analysis were natural killer (NK) cell-mediated cytotoxicity, immunoregulatory actions, leukocyte activity, and IL-12 pathway (Figure 1).

In contrast, there were no significant gene expression changes in ME/CFS patients when comparing T1 to T0. 

#### Cell Type Abundance Changes between Maximal Exertion (T1) and Baseline before Exercise Challenge (T0) 

Abundances of cell types were estimated using normalized gene counts. Naive CD4+ T cells were decreased in HCs, while NK cells were significantly increased at T1 time point compared to T0 (Table 2). There were no significant changes in immune cell types found in ME/CFS patients between maximal exertion and baseline before exercise challenge (Table 2). 

### 2.3. Transcriptomic Changes between 4 h after Maximal Exertion (T2) and Maximal Exertion (T1)

We investigated gene expression in PBMCs from ME/CFS patients and HCs between time points T2 (4 h after maximal exertion) and T1 (maximal exertion). In HCs, 831 genes had significant changes in their level of expression, with 542 being underexpressed (Appendix A). ME/CFS patients had 1277 genes that were differentially expressed, with 892 of them underexpressed (Appendix A). 

The functional analysis of DEGs resulted in several clusters that were significantly affected in ME/CFS patients when compared to HCs. They included herpes virus 1 infection, cytokine signaling, cellular response to cytokine stimulus, positive regulation of cytokine production, cellular response to stress, and regulation of cellular response to stress. In HCs, leukocyte activation and immune response-regulating signaling were more significantly affected, as compared with ME/CFS patients (Figure 2).

#### Cell Type Abundance Changes between 4 h after Maximal Exertion (T2) and Maximal Exertion (T1)

In ME/CFS patients, naive CD4+ T cells were significantly elevated, while dendritic cells and eosinophils were significantly decreased (Table 3). HCs showed significant decreases in memory B cells, CD8+ T cells, NK cells, and eosinophils. In addition, HCs had significant increases in naive and memory CD4+ T cells, as well as activated mast cells (Table 3). 

### 2.4. NanoString Validation

Approximately 90 percent of the DEGs identified by RNA-Seq were validated by Nanostring Technologies nCounter (Appendix A). Genes with discrepancies between Nanostring and RNA-Seq were not used in functional analysis. 

## 3. Discussion

Historically, most ME/CFS “omic” studies have investigated participants while at rest. By incorporating an exercise challenge, our study design allowed us to analyze temporal transcriptomic changes in PBMCs of ME/CFS patients in response to exercise and during recovery and to highlight unique pathways that are altered in response to physical stress. Studies have documented that the prevalence and symptom severity of ME/CFS is higher in women than in men [31]. Additionally, prior research has demonstrated sex differences between ME/CFS patients on a molecular level [13,25,32,33,34]. For these reasons, in this study we have chosen to focus on female ME/CFS patients only.

### 3.1. Transcriptomic Changes between Maximal Exertion (T1) and Baseline before Exercise Challenge (T0)

Our results demonstrated no significant differences in the gene expression of circulating immune cells in female ME/CFS patients at maximal exertion (T1), as compared to baseline before exercise challenge (T0). Recent work from our group has also supported this finding. We showed that female ME/CFS patients do not show significant changes in the levels of microRNAs in their PBMCs between baseline before exercise challenge and the point of maximal exertion [13]. MicroRNAs are small, non-coding RNA molecules that regulate gene expression through translation repression [35]. Since the regulation of transcriptomes in ME/CFS patients is unchanged between baseline and point of maximal exertion, it is not surprising that gene expression remains relatively stagnant during this time. This finding is noteworthy because ME/CFS is characterized by chronic fatigue [36]. At any time, ME/CFS patients have feelings of exhaustion [37]. Therefore, when ME/CFS patients are subjected to an exercise challenge, transcriptomic changes in their PBMCs are not significant enough to pass criteria for differential expression. 

In contrast, HCs had multiple genes and functional pathways that were altered in their PBMCs as they went from baseline before exercise challenge (T0) to maximal exertion (T1). It is noteworthy that most genes were overexpressed. Currently, the reason for this phenomenon remains unclear. One possible explanation is that an immune response is activated as a result of tissue damage due to exercise. For example, we identified changes in genes that are related to lymphocyte differentiation, signaling, and fate (MAL and SLC16A10). 

The analysis of these gene networks in HCs showed that the most significant changes were in pathways related to NK cell-mediated cytotoxicity (Figure 1). NK cells are effector lymphocytes that serve as part of the innate immune system. NK cells kill virally infected or malignantly transformed cells via the direct release of lytic granules or by inducing death receptor-mediated apoptosis via the expression of the Fas ligand [38]. Our data indicate that, in response to an exercise challenge, HCs can exert significant transcriptomic changes in their NK cells to meet the challenge of a stressor. 

In addition, CIBERSORTx results show that NK cells were significantly increased in HCs at T1 (maximal exertion) compared to T0 (baseline before exercise challenge). On the other hand, numerous studies support reduced NK cell activity in patients with ME/CFS [39,40,41]. These abnormal NK cells may not be able to properly alter their gene expression, making ME/CFS patients less able to react to a physical stressor. Cell type abundance analysis showed that there were no changes in NK cells in ME/CFS patients at maximal exertion compared to baseline before exercise challenge (Table 2). These data suggest dysfunction of the innate immune system may be contributing to PEM in ME/CFS patients. 

Another pathway that showed significant change from baseline before an exercise challenge (T0) to maximal exertion (T1) in HCs was immunoregulatory interactions between lymphoid and non-lymphoid cells. A variety of receptors and cell adhesion molecules play a role in modifying the response of lymphocytes to antigens that are present on somatic cells [42]. This critical surveillance system alerts the immune system to internal damage. These changes in HCs suggest that significant interactions are taking place between lymphoid cells and non-lymphoid cells to coordinate a proper response to an exercise challenge. These signals may be defective in patients with ME/CFS, as evidenced by studies showing extensive morphological alterations in the cellular phenotype of their immune cells [43]. Specifically, ME/CFS patients exhibit alterations in NK and T cell receptors, which may disrupt communication between immune cells and somatic cells [44]. 

Furthermore, HCs had distinct changes in the IL-12 pathway that play a central role in promoting type 1 T-helper cell responses [45]. Although HCs can modify the expression of this gene network in response to a physical stressor, a recent study documented that elevated levels of IL-12 correlated with the development of ME/CFS after infection with mononucleosis [46]. These findings may suggest that an inflammatory response is still active in ME/CFS patients, despite a threat being eliminated. This indicates that regulators of the IL-12 pathway may be dysfunctional in ME/CFS patients when they are exposed to a stressor, such as a virus or physical exertion, whereas HCs are able to regulate the expression of this gene. 

Overall, these data suggest that HCs can significantly alter the transcriptomics of their immune cells to meet the demands of a physical stressor, while ME/CFS patients cannot promote any changes in gene expression that are significant enough to pass criteria for differential expression. 

### 3.2. Transcriptomic Changes between 4 h after Maximal Exertion (T2) and Maximal Exertion (T1)

We compared the similarities and differences in functional pathways altered in PBMCs between female ME/CFS patients and HCs at 4 h after maximal exertion (T2) and maximal exertion (T1).

Both cohorts show similar alterations in pathways related to orexin and cellular migration (Figure 2). These cellular functions may only be slightly affected by ME/CFS and are, therefore, not critical for analyzing when trying to find further avenues for diagnostic and treatment options.

We found many transcriptomic changes and functional pathway changes that were different between ME/CFS patients and HC between 4 h after maximal exertion (T2) and maximal exertion (T1). Specifically, genes related to cytokine signaling were overexpressed (CISH and CCR2), while genes related to helping protect cells from the adverse effects of stress were downregulated (HSPA1B and DDIT3). 

Functional pathway analysis showed that gene networks responsible for modifying cellular activity in response to bacteria and viruses (“hsa05186”: Herpes Simplex Virus 1 Infection and “GO:0002237”: Response to molecules of bacterial origin) showed significant changes. The onset of most ME/CFS cases begins after exposure to an infection, most commonly viral infections, especially those from the herpesvirus family [47]. Herpes viruses are known for their ability to establish lifelong infections. For example, after an initial HSV1 infection, the virus is transported along axons and takes up life-time residency in nerve cells. The reactivation of these viruses is then possible, often induced by systemic stress, such as transient hyperthermia [48]. Studies have found that the reactivation of herpes viruses in ME/CFS patients drives mitochondrial remodeling that pushes cells toward a hypometabolic state and prevents energy from being available for cellular functions, such as growth and repair [49]. This data suggests that stress (such as an exercise challenge) may facilitate the reactivation of latent herpes viruses in ME/CFS patients, which may severely compromise cellular energy metabolism and potentially lead to PEM. 

Another set of pathways that were significantly affected in ME/CFS patients but not in HCs between 4 h after maximal exertion (T2) and maximal exertion (T1), corresponding with how cells respond to stress (Reactome “R-HSA-2262752: Cellular responses to stress”, Gene Ontologies “GO:1902532: negative regulation of intracellular signal transduction” and “GO:0080135: regulation of cellular response to stress”). The ability of cells to modulate molecular processes in response to external stressors is essential to the maintenance of homeostasis [50]. In this study, our model of stress was exercise, which produces free radicals as aerobic cellular metabolism increases to provide energy to skeletal muscles. These reactive oxygen species must be reduced by antioxidant defenses to prevent damage to DNA, lipids, and proteins. This protective system appears to be dysregulated in patients with ME/CFS. A recent report showed a decreased production of the antioxidant nitric oxide (NO) in endothelial cells exposed to plasma from ME/CFS patients [51]. This correlates with studies that have shown increased levels of oxidative and nitrosative stress in ME/CFS [52]. These data suggest that after significant exertion, ME/CFS patients are unable to mount the proper defenses to combat cellular stress leaving their immune cells vulnerable to apoptosis. 

Other pathways that were significantly changed in ME/CFS patients as they recovered from an exercise challenge were related to cytokines (Figure 2). One such gene network was the positive regulation of cytokine production (“GO:0001819: positive regulation of cytokine production”). Four hours after maximal exertion, ME/CFS patients are continuing to increase the production of cytokines. This dysfunctional regulation process is supported by several studies showing altered levels of cytokines in the blood of patients with ME/CFS [53,54,55,56]. The other pathway that was significantly affected was the cellular response to cytokines (“GO:0071345: cellular response to cytokine stimulus”). Although cytokines are known to promote immune activity, ME/CFS patients have defective responses to these immune molecules. Specifically, one research group showed that in response to cytokines, ME/CFS patients have reduced metabolism in their CD8+ T cells [57]. These studies suggest that in response to a physical stressor, immune cells in patients with ME/CFS are not only associated with the dysregulated secretion of cytokines but also linked to a dysfunctional response to them. 

At 4 h after maximal exertion (T2) compared to maximal exertion (T1), HCs showed changes in lymphocyte gene expression that were significantly different from those of ME/CFS patients. Genes that were downregulated were related to inflammatory signaling in the lymphocytes, including AKAP1, SMAD7, and SPON2. Specifically, inflammatory pathways responsible for lymphocyte activation and regulation were modified (“GO: 0002764”: immune response-regulating signaling pathways and “GO:0045321”: leukocyte activation). These gene networks regulate the coordinated synthesis of molecular mediators that work together to perpetuate an inflammatory reaction. In response to tissue damage from exercise, HCs have a synchronous activation of their immune cells that mediate damage and then repair. Meanwhile, ME/CFS patients are characterized by constant low-grade inflammation [58].

By not properly reducing inflammation, ME/CFS patients have altered lymphocyte cell populations. Recent studies have found increased proportions of mucosal-associated invariant T cells, as well as effector CD8+ T cells in patients with ME/CFS [59]. Meanwhile, the proportions of CD8+ T cells were significantly decreased in HCs during the recovery period (Table 3). In order to recover from a physical stressor, HCs may be able to inactivate their immune cells and reduce inflammation, whereas these pathways are dysregulated in patients with ME/CFS leading to constant low-grade inflammation. 

In prior studies, it was shown that during the recovery period after exercise, healthy individuals undergo an increase in circulating monocytes and neutrophils, as well as a decrease in lymphocytes. It has been hypothesized that this lymphopenia is caused by an extravasation of these cells out of circulation and into the periphery [60]. In the healthy population, recovery from exercise/stress is likely dependent on this described immune response, as well as the normalization of immune cell counts and distributions throughout body tissues. Table 3 shows that cell counts change differently in ME/CFS patients and HC during recovery from exercise, suggesting that ME/CFS patients may have dysfunctional recruitment and mobilization of these immune cells.

Taken together, these data suggest that between 4 h after maximal exertion (T2) and maximal exertion (T1), ME/CFS patients have significant transcriptomic changes in pathways related to viruses, cellular stress, and cytokine production, while HCs alter functional pathways involving lymphocyte regulation. These gene expression changes may indicate that dysregulated immune responses are contributing to the PEM that characterizes this disease.

### 3.3. Potential Epigenetic Dysregulation of DEGs and Functional Pathways 

MicroRNAs are small non-coding RNAs that have been shown to regulate gene expression. In most cases, microRNAs interact with the 3′UTR of target mRNAs to suppress translation. In some cases, microRNAs’ interaction with target messenger RNA (mRNA) lead to mRNA degradation [61].

Previously, our group analyzed the differential expression of microRNAs in ME/CFS patients during and after an exercise challenge [13]. We found that hsa-miR-146a-5p was overexpressed in female ME/CFS patients but not in HC between 4 h after maximal exertion (T2) and maximal exertion (T1). This microRNA has been shown to regulate the expression of the gene TRAF6 [62]. It is noteworthy to mention that in the current study, TRAF6 was significantly downregulated in ME/CFS patients (but not HC) at 4 h post-exercise challenge, as compared to maximal exertion (Appendix A). TRAF6 has been identified as a signal transducer important for the regulation of inflammation and immunity [63]. 

In addition, the microRNA hsa-miR-146a-5p has been shown to regulate the gene YES1 [64]. In our current study, we found that YES1 was significantly down regulated in ME/CFS patients (but not HC) at 4 h post-exercise challenge compared to maximal exertion (Appendix A). Dysfunction in protein kinase genes, such as YES, has been found in severe cases of ME/CFS and may contribute to impairments in NK cell intracellular signaling and effector functions [65]. These potential correlations between dysregulated epigenetics leading to gene expression abnormalities should be an avenue for the future research studies in order to elucidate the mechanisms behind the DEGs and functional pathways identified in this study.

### 3.4. Limitations

In prior studies, both symptomatic and molecular differences have been elucidated between men and women diagnosed with ME/CFS [13,32,33,34]. As such, our conclusions are limited to female patients only. Further studies comparing transcriptomic sex-differences in ME/CFS are required in order to make more generalized conclusions about the disease.

## 4. Materials and Methods 

### 4.1. Cohort

A community-based cross-sectional study included 20 clinically diagnosed female ME/CFS subjects and 20 female healthy controls (HC) matched for age (±5 years) and BMI (±2 units). All individuals with ME/CFS and HCs were recruited from the Miami/Fort-Lauderdale area as a part of a large ongoing study by the Institute for Neuroimmune Medicine. All subjects signed informed consent for inclusion before they participated in the study. The study was conducted in accordance with the Declaration of Helsinki, and the protocol was approved by the institutional review board of Nova Southeastern University (2016-2-NSU). All ME/CFS subjects met the 1996 CDC/Fukuda and 2003 Canadian Case definitions for ME/CFS. Patients were excluded from the study if they had a history of active smoking or alcohol, diabetes, immunodeficiency, cardiovascular disease, stroke, autoimmunity, malignancy, or systemic infection within 2 weeks of blood collection. The short-form 36-item health survey (SF-36) questionnaire [28] was used to compare individuals with ME/CFS and HC. This questionnaire includes “Physical Health” (which is comprised of physical functioning, physical role functioning, bodily pain, and general health perception) and “Mental Health” (which includes vitality, social functioning, emotional role, and mental health). Each of these eight domains is transformed into a 0–100 point scale, with each question being weighted equally. Higher scores on this scale indicate lower levels of disability. In other words, a score of 0 is demonstrative of maximal disability and a score of 100 is demonstrative of no disability in that particular domain [65]. Subjects also completed a gynecologic questionnaire to ensure blood collection occurred during the first two weeks of their menstrual cycle. 

After a uniform breakfast (yogurt and banana), the subjects were given 30 min to digest while resting in comfortable position in reclining chairs (T0, baseline before exercise challenge). Immediately after, an exercise challenge was performed utilizing a standard maximal graded exercise test (GXT) according to McArdle’s protocol [66]. This protocol was used as part of larger ongoing studies of biological mechanisms underlying neuroimmune diseases. Each participant pedaled at 60 W for 2 min, followed by an increase of 30 W every 2 min until they reached their maximal exertion. The second blood draw was conducted upon reaching this point of maximal exertion (T1) and the third blood draw was performed at 4 h after maximal exertion (T2—recovery).

### 4.2. PBMC Isolation and RNA Extraction

Up to 8 mL of whole blood was collected from each participant into K_2_EDTA tubes and diluted at 1:1 (*v*/*v*) ratio in PBS within 2 h. As described before [13], this solution was layered on top of Ficoll-Paque Premium (GE Healthcare, Chicago, IL) and separated by density centrifugation at 500× *g* for 30 min (20 °C with brakes off). The PBMC layer was isolated, washed with PBS, resuspended into one volume of red blood cell lysis buffer, kept on ice for 5 min and then centrifuged at 500× *g* for 10 min. The PBMC pellets were resuspended in freezing medium with aliquots of 10^7^ cells/mL frozen in liquid nitrogen until use. RNAzol (Molecular Research Center, Cincinnati, OH) was used to extract total RNA and the quality was assessed using Agilent TapeStation 4200 (Agilent Technologies, Santa Clara, CA). All RNA samples had an RNA integrity number above seven. 

### 4.3. RNA Sequencing 

Total RNA (500 ng) was submitted to the Center of Genome Technology (CGT) at the University of Miami for RNA sequencing. Libraries were generated using TruSeq Stranded Total RNA library Prep Kit (Illumina Inc., San Diego, CA) with paired-end sequencing reading length of 150 nucleotides. CGT staff used the Illumina RNA-Seq pipeline to assess genomic coverage, percent alignment, and nucleotide quality for quality control assessment. Raw sequencing data were then transformed to the fastq format. 

### 4.4. RNA-Seq Analysis

Raw reads were mapped to the reference human genome (GRCh38, release 86) using GSNAP [67], HISAT2 [68], and STAR [69] software. Reads that were aligned by GSNAP and HISAT2 were counted using HTSeq software [70], while alignment by STAR was run with the option “--quantMode Transcriptome SAM”. Counts from HTSeq and STAR were uploaded into Bioconductor/R package DESeq2 [71] and tested for differential gene expression. Only genes with raw counts of 50 or above in all samples in at least one group (ME/CFS patients at T0, ME/CFS patients at T1, ME/CFS patients at T2, HC at T0, HC at T1 and HC at T2) were used for the differential analysis by DESeq2. DEGs were then selected based on the following criteria: (1) either a fold change (FC) > 1.5 and false discovery rate (FDR) < 0.10 in 1 out of 3 aligners and FC > 1.4 and FDR < 0.15 in the remaining 2 aligners OR (2) a FC > 1.5 and FDR < 0.10 in at least 2 out of 3 aligners. 

DEGs were then uploaded to Metascape [29] and an Express Analysis was performed using default criteria (*p*-value < 0.01, a minimum count of three, and an enrichment factor > 1.5). Cytoscape [72] was used for visualizing networks. 

To analyze possible changes in the lymphocytes subtypes we used CIBERSORTx [30] software utilizing normalized gene counts. The default LM22 leukocyte gene signature matrix was used as a reference. Quantile normalization was disabled, and the number of permutations was set to 1000. ANOVA was used to find statistically significant differences in cell counts, and *p* value of 0.05 was used as a cut off.

### 4.5. Validation of RNA-Seq Results

The Nanostring nCounter platform was used to validate the RNA-Seq results using a custom panel. Total RNA (100 ng) was used for processing and hybridization. Sample clean-up and counting were performed according to the manufacturer’s instructions. NanoString nSolver v.4 software was used to analyze raw counts. We calculated geometric means of negative controls plus two standard deviations for all samples and counts below this threshold value were excluded from normalization and analysis. After this, all steps were performed according to manufacturers’ instructions.

## 5. Conclusions

Overall, our results indicate that female ME/CFS patients respond differently to an exercise challenge that stimulates PEM, as compared to female HCs. Under maximum stress, ME/CFS patients are unable to facilitate transcriptomic changes in the cells of their immune system that would allow them to permit recovery. Meanwhile, in response to stress, HCs make several transcriptomic changes related to the integrity and function of their immune cells. During recovery, ME/CFS immune cells have dysfunctional cytokine signaling networks and are vulnerable to cell death due to poor defense systems and dysregulated epigenetic regulation of apoptotic pathways, whereas HCs regulate their lymphocytes and inactivate the inflammatory response. The findings presented in our study will help to advance our understanding of the pathways and genes that are relevant to PEM and fatigue tied to ME/CFS, with the goal of identifying targets to improve diagnosis and identify more targeted therapeutic options for female ME/CFS patients. 

## Figures and Tables

**Figure 1 ijms-24-02698-f001:**
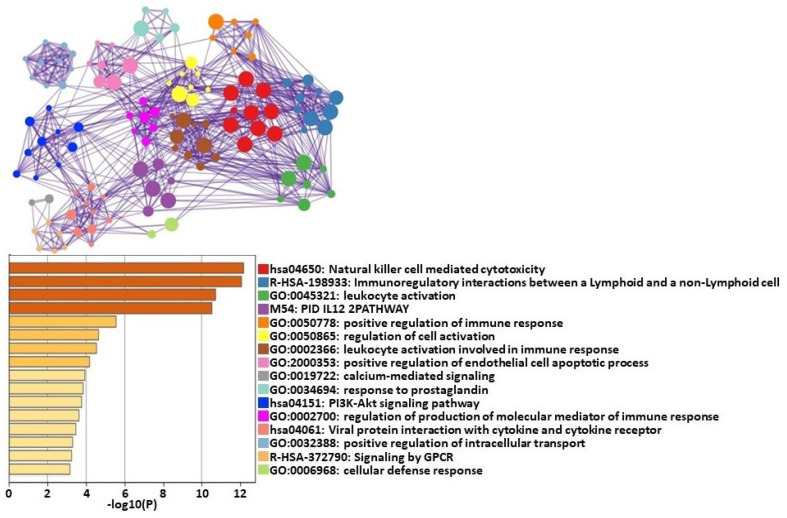
Metascape [29] express analysis of all 102 differentially expressed genes between HCs at T0 and T1. Each node represents an enriched term. These terms are colored by cluster ID to indicate which cluster they belong to. Cutoff values included a *p*-value of <0.01, a minimum count of 3, and an enrichment factor > 1.5. GO biological processes, KEGG pathways, Reactome gene sets, CORUM complexes, and canonical pathways from MSigDB were included in the search.

**Figure 2 ijms-24-02698-f002:**
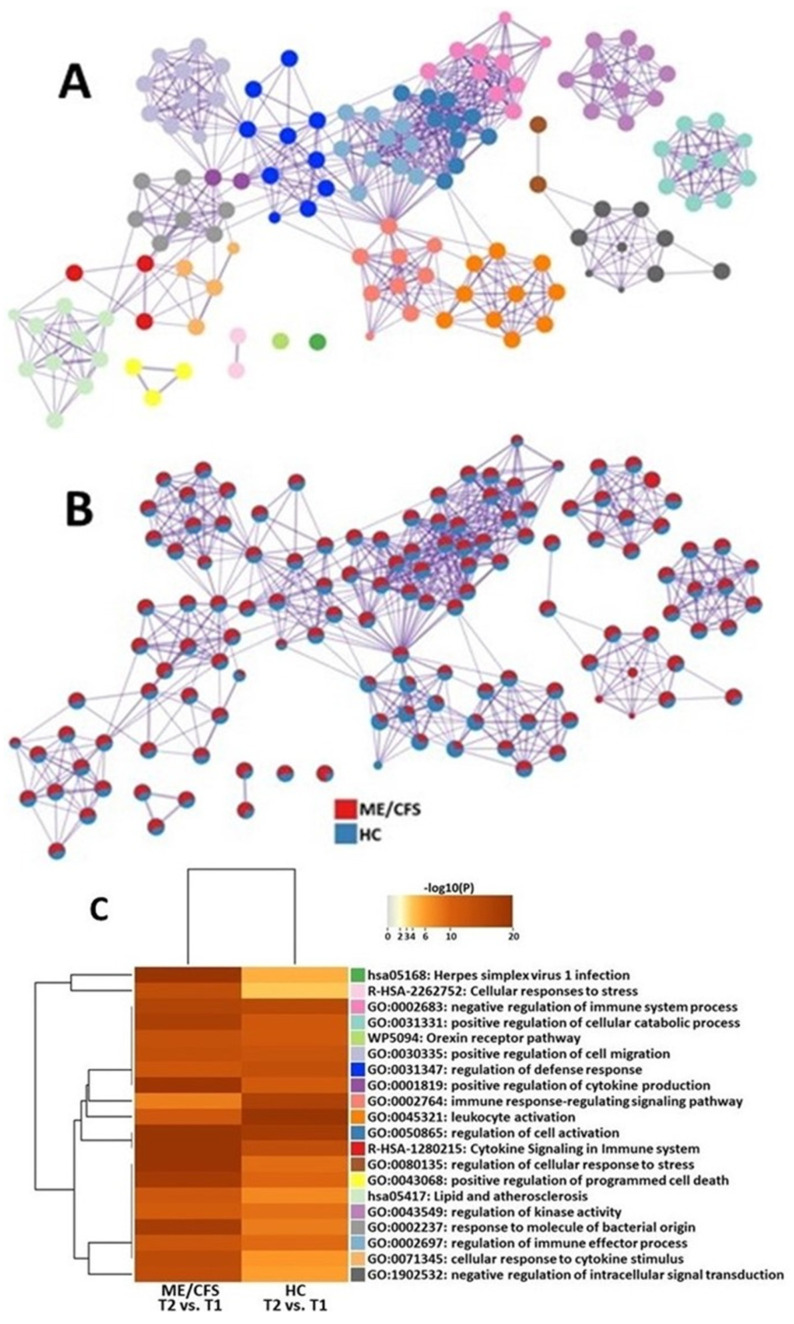
Metascape [29] express analysis of differentially expressed genes in ME/CFS patients and HCs between T1 and T2. (**A**) Each node represents an enriched term, and colors of the nodes are explained in (**C**). The larger the representation of the color the more differentially expressed in each cohort. Cutoff values included a *p*-value of <0.01, a minimum count of 3, and an enrichment factor of >1.5. GO biological processes, KEGG pathways, Reactome gene sets, CORUM complexes, and canonical pathways from MsigDB were included in the search. (**B**) Nodes are colored by rations ME/CFS to HC. The red color indicates functional pathways in ME/CFS patients while the blue color indicates functional pathways in HCs. (**C**) Heatmap of enriched terms across lists of DEGs between T2 and T1 in PBMC of ME/CFS patients and HCs.

**Table 1 ijms-24-02698-t001:** Demographic information comparing ME/CFS and HC. Data are shown as mean ± standard error of mean, *—*p* ≤ 0.05, Student *t*-test.

		ME/CFS Cases	Healthy Controls	*p*-Value
	Age (years)	46.8 ± 2.14	46.4 ± 2.06	0.883
	BMI (kg/m^2^)	26.6 ± 1.17	26.5 ± 1.12	0.913
Physical Health				
	Physical Functioning	39.2 ± 5.55	96.3 ± 1.47	<0.001 *
	Role Physical	15.3 ± 6.44	92.1 ± 5.43	<0.001 *
	Bodily Pain	39.1 ± 6.67	89.6 ± 2.85	<0.001 *
	General Health	26.0 ± 4.57	77.2 ± 4.20	<0.001 *
Mental Health				
	Vitality	23.2 ± 4.17	61.5 ± 6.46	<0.001 *
	Social Functioning	38.2 ± 5.46	90.1 ± 3.77	<0.001 *
	Role Emotional	64.8 ± 10.25	87.7 ± 6.35	0.068
	Mental Health	44.6 ± 4.35	77.8 ± 4.25	<0.001 *

**Table 2 ijms-24-02698-t002:** Cell type abundance data from CIBERSORTx [30] analysis between T1 (maximal exertion) and T0 (resting state) in healthy controls. * indicates statistically significant changes in cell counts.

T1 v. T0 in ME/CFS Patients		
Cell Type	*p*-Value	Fold Change
CD4+ T cells naive	0.620	−1.123
CD4+ T cells memory resting	0.621	1.048
CD4+ T cells memory activated	0.589	−1.302
NK cells	0.189	1.234
**T1 v. T0 in Healthy Controls**		
**Cell Type**	***p*-Value**	**Fold Change**
CD4+ T cells naive	0.043 *	−1.983
CD4+ T cells memory resting	0.080	−1.190
CD4+ T cells memory activated	0.083	−3.828
NK cells	0.001 *	1.637

**Table 3 ijms-24-02698-t003:** Cell type abundance data from CIBERSORTx [30] analysis between T2 (4 h after maximal exertion) and T1 (maximal exertion). * indicates statistically significant changes in cell counts.

T2 v. T1 in ME/CFS Patients		
Cell Type	*p*-Value	Fold Change
B cells naïve	0.077	2.534
CD4+ T cnaïvenaive	0.022 *	1.572
Dendritic cells	0.001 *	−2.013
Eosinophils	<0.001 *	−2.938
**T2 v. T1 in Healthy Controls**		
**Cell Type**	***p*-Value**	**Fold Change**
B cells memory	0.030 *	−1.244
CD8+ T cells	0.001 *	−13.444
CD4naïveells naive	<0.001 *	3.335
CD4+ T cells memory activated	0.017 *	4.718
NK cells	<0.001 *	−1.869
Mast cells activated	0.006 *	2.789
Eosinophils	0.009 *	−2.382

## Data Availability

The raw sequencing data presented in this study are available on request from the corresponding author. The data are not publicly available due to the use of this data in another study. Data will be publicly available upon the publication of the second study.

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
