# Peer review of "Stress-Induced Transcriptomic Changes in Females with Myalgic Encephalomyelitis/Chronic Fatigue Syndrome Reveal Disrupted Immune Signatures"

_ijms, 2023, doi:10.3390/ijms24032698_

Round 1

Reviewer 1 Report

This will be an interesting article for the ME/CFS research community and adds valuable information to the molecular evidence available for differences between ME/CFS patients and healthy controls. There are other molecular studies with other technologies that also report important differences and perhaps brief referencing of those in the introduction would have been helpful so this study is seen in a wider context. It takes a stress model of exercise (and a core symptom -post exertional malaise) and maps what happens in both healthy controls and ME/CFS patients - in this case a female cohort. From existing data ( eg increase in cytokines rather than decrease) it would be expected there would be differences between the ME/CFS and healthy controls groups in the recovery phase as was found, but the surprising finding is the absence of any transcriptional activity changes at maximum exercise compared with the healthy controls. What prevents an immediate  molecular response to the extra stress in the ME/CFS patients who are so sensitive to stress? 

Because the protocol of this study is unique all of the well documented differences are interesting to add to the available data that will give a complete understanding of the dysfunctional physiology and abnormal molecular homeostasis in ME/CFS .The study complements information from other molecular technologies and thereby makes an important contribution to understanding the pathophysiology of ME/CFS. The findings were clearly presented, easy to follow and well justified.

My specific comments and minor criticisms are listed below:

Abstract:(1) "there are no well-established biosignatures tied to ME/CFS to target treatment more effectively"

This may be strictly true but a little misleading -there are biosignatures like ' neuroinflammation', ' disturbed mitochondrial protein production in immune cells, and 'epigenetic changes reflecting changed gene expression' documented that suggest targeted treatments - for example anti- neuroinflammatory agents, antioxidants, etc. This study provided valuable information but similarly does not do more that indicate areas where targeted treatment might be possible. The critical thing is there have been no clinical trials for those treatments that have been suggestive of being helpful- hopefully the intensive research activity now on Long COVID might address this deficit.

2. In the introduction it would be helpful to give brief mention of these other 'omic' technologies and molecular changes observed that form a complement to this study. The introduction gives an excellent broad overview but it would be good to have a somewhat more in depth holistic focus from the collective studies aimed at addressing  the poorly understood pathophysiology -as this study does- so that the stress model findings are indicative and informative of what is happening day to day in the lives of ME/CFS patients in their inability to cope with normal day to day stresses.

3. Maximal exertion will vary with each individual and i am assuming the ME/CFS patients reached that at much lower load. How much do you think that might have influenced the finding that no transcriptional changes in the ME/CFS  were found ie the physical stress while effecting cardiac physiology may not have been significant enough to trigger immediate signals to the genome.  A comment would be worthwhile on this.I am not sure I understand the comment 'they were at maximal exertion at base line' as they did complete an exercise protocol. Stress can send ME/CFS patients into a relapse from their normal restricted level of health so fatigue gets worse just as it does post exercise. Hence I am not convinced by the argument that because they are exhausted at base line they are at maximum exertion  - if so they would have been incapable of doing the study as acute homebound/bedridden patients would have been.

4. Perhaps helpful in the text to describe the categories 'Physical health' and 'Mental health' in Table 1 derived from  the 36 question survey so that readers  who are not familiar with it  get a better feel of what it is measuring and waht the different numbers mean (ie point range, how do number rating relate to subjective rating etc).

5. How many total  transcripts were analysed in the two groups? Can you comment further on why virtually all differential gene expression in the HC T1 vs T0 was the result of downregulation of expression. It is interesting by contrast in the T2 vs T1 most genes were upregulated. 

Author Response

  1. "there are no well-established biosignatures tied to ME/CFS to target treatment more effectively"

This may be strictly true but a little misleading -there are biosignatures like ' neuroinflammation', ' disturbed mitochondrial protein production in immune cells, and 'epigenetic changes reflecting changed gene expression' documented that suggest targeted treatments - for example anti- neuroinflammatory agents, antioxidants, etc. This study provided valuable information but similarly does not do more that indicate areas where targeted treatment might be possible. The critical thing is there have been no clinical trials for those treatments that have been suggestive of being helpful- hopefully the intensive research activity now on Long COVID might address this deficit.

We greatly appreciate your time and insight in providing commentary around the word utilization of well-established biosignatures. You make a very valid point. As such, we are taking into consideration your comments, and have removed the sentence from the abstract addressing this. In addition, we clarified that our study details transcriptomic changes that may be used for further research focusing on the identification of biomarkers, rather than saying that we identified unique biomarkers. We believe that removes any confusion around this commentary, as we did not mean to be misleading.

  1. In the introduction it would be helpful to give brief mention of these other 'omic' technologies and molecular changes observed that form a complement to this study. The introduction gives an excellent broad overview but it would be good to have a some what more in depth holistic focus from the collective studies aimed at addressing the poorly understood pathophysiology -as this study does- so that the stress model findings are indicative and informative of what is happening day to day in the lives of ME/CFS patients in their inability to cope with normal day to day stresses.

Thank you very much for your valuable feedback. In response to this thoughtful commentary, we have added a section to the introduction detailing prior ‘omic’ studies completed involving ME/CFS patients. Specifically, we added information and included studies that evaluated changes associated with metabolomics, proteomics, epigenetics, and transcriptomics among ME/CFS patients at rest. In addition, we also added information and studies that investigated changes in metabolomics and transcriptomics among ME/CFS patients in response to exercise. Furthermore, we provided insight into how our study differs from previously published studies, and how it adds to the body of knowledge and addresses unmet needs in the field. 

  1. Maximal exertion will vary with each individual and i am assuming the ME/CFS patients reached that at much lower load. How much do you think that might have influenced the finding that no transcriptional changes in the ME/CFS were found ie the physical stress while effecting cardiac physiology may not have been significant enough to trigger immediate signals to the genome. A comment would be worthwhile on this. I am not sure I understand the comment 'they were at maximal exertion at baseline' as they did complete an exercise protocol. Stress can send ME/CFS patients into a relapse from their normal restricted level of health so fatigue gets worse just as it does post exercise. Hence I am not convinced by the argument that because they are exhausted at base line they are at maximum exertion - if so they would have been incapable of doing the study as acute homebound/bedridden patients would have been.

We appreciate the thoughtfulness and rigor in your comments provided. In taking into consideration these comments, in section 3.1., we removed the statement that “ME/CFS patients were at maximal exertion at baseline.” Instead, we have clarified that compared to healthy controls (HC), ME/CFS patients have some level of fatigue at baseline. While they were still able to complete an exercise challenge and therefore not at “maximal exertion,” the changes observed in ME/CFS transcriptomes were not statistically significant enough to meet our criteria for differentially expressed genes.

  1. Perhaps helpful in the text to describe the categories 'Physical health' and 'Mental health' in Table 1 derived from the 36 question survey so that readers who are not familiar with it get a better feel of what it is measuring and what the different numbers mean (ie point range, how do number rating relate to subjective rating etc).

We appreciate your commentary on this, and believe it would be helpful to the reader and other investigators. As such, in the legend of Table 1, we have detailed where the values come from and what higher/lower scores indicate in terms of disability. In addition, in section 2.1., we added an explanation of what the scores mean in a broader context. Furthermore, in section 4.1, we explained which categories of the questionnaire correspond to mental health and physical health, how the scores are calculated based on the questions answered, and what the scores mean.

  1. How many total transcripts were analyzed in the two groups? Can you comment further on why virtually all differential gene expression in the HC T1 vs T0 was the result of downregulation of expression. It is interesting by contrast in the T2 vs T1 most genes were upregulated.

Thank you for this commentary, as we agree with you that this warrants further explanation. In response, in section 4.4, we clarified the total number of transcripts analyzed in the two groups. In addition, in section 3.2., we commented on the trends of over- and under-expression seen. Meanwhile, the exact cause of why this is occurring remains unclear, though we have provided a hypothesis (mobilization of the immune system in response to tissue damage from exercise) and how this is related to our data. 

Reviewer 2 Report

The authors reported on the reaction of the immune system to an effort in patients with Myalgic Encephalomyelitis/Chronic Fatigue Syndrome (ME/CFS). 20 female patients were submitted to an effort and their immune system transcriptome was compared to this of age-matched female healthy controls. The transcriptome from PBMCs was analyzed at the time, directly after the effort and after recovery.

The study is interesting but there are some conceptual issues.

Major remarks:

11)      According to the study protocol, patients sat 30 minutes long before being submitted to the effort. As sitting can be a stress cause in some ME/CFS patients, there should be a “baseline assessment” of the immune system for patients and healthy controls at the time of the effort. Without this comparison, it is difficult to interpret the differences observed by the authors.

22)      The conception of the discussion, which presents some results of the study, is not clear. Moreover, Table 4 presents a correlation between the transcription levels of certain genes and regulating miRNA without a precise idea, why these genes are underlined and, which role they could play in the physical recovery of the patients.

33)      There should be a clarification about the influence of the immune system on the physical recovery of ME/CFS patients and/or healthy controls. How does the immune system contribute to the physical recovery in healthy individuals? Are there some hypotheses about the influence of the immune deficiency of ME/CFS patients on their lack of physical recovery?

Minor remarks:

11)      Could the authors be more precise and define why they qualify the effort the patients submit to as “modeled stress”? Can the authors precise why they chose to submit the study patients to such a strong physical test, when other groups used tests that are milder?

22)      There should be a mention in the abstract and the introduction that only female patients were considered, which could induce a gender bias in the results. The authors could add a part on the influence of gender in ME/CFS and the immune system in the discussion to illustrate this issue.

Author Response

Major remarks:

  1. According to the study protocol, patients sat 30 minutes long before being submitted to the effort. As sitting can be a stress cause in some ME/CFS patients, there should be a “baseline assessment” of the immune system for patients and healthy controls at the time of the effort. Without this comparison, it is difficult to interpret the differences observed by the authors.

We greatly appreciate the thoughtfulness in this commentary. In response, in section 4.1., we further explained that after breakfast and before exercise, participants relaxed in reclining chairs in a comfortable position. In addition, in section 2.1., we have clarified that all participants in the study were required to be able to complete an exercise challenge.

2.The conception of the discussion, which presents some results of the study, is not clear. Moreover, Table 4 presents a correlation between the transcription levels of certain genes and regulating miRNA without a precise idea, why these genes are underlined and, which role they could play in the physical recovery of the patients.

Thank you very much for this valuable feedback. We agree with your comments. After careful consideration of your commentary, we removed the miRNA table and miRNA results section (section 2.6.) from the manuscript. We believe that this will clarify the conception of the discussion and allow for a closer focus on the results of this current study. In an effort to correlate our prior miRNA data with the transcriptomic changes observed in this current study (without compromising clarity of the manuscript), we only discussed how hsa-miR-146a-5p (that changed only in ME/CFS patients but not HCs) and some corresponding DEGs relate to the recovery of patients (section 3.6.).

3.There should be a clarification about the influence of the immune system on the physical recovery of ME/CFS patients and/or healthy controls. How does the immune system contribute to the physical recovery in healthy individuals? Are there some hypotheses about the influence of the immune deficiency of ME/CFS patients on their lack of physical recovery?

We greatly appreciate your thoughtfulness in the review of our manuscript and wish to address your commentary above. In response, in section 3.4., we provided a description, based on prior studies, of how a healthy individual’s immune system will respond following exercise.  Specially, we added a description in the manuscript of how there is an increase in neutrophils and monocytes, and a relative lymphopenia presumably from extravasation of these cells into peripheral tissues [Peake JM, Neubauer O, Walsh NP, Simpson RJ. Recovery of the immune system after exercise. J Appl Physiol (1985). 2017;122(5):1077-87]. This additional commentary also connected these prior findings with the cell type abundances, differentially expressed genes, and pathway analysis data acquired from our study. We also provided a potential hypothesis regarding the underlying mechanisms, in which immune deficiency seen in patients with ME/CFS may impact recovery as compared to HCs. 

Minor remarks:

1.Could the authors be more precise and define why they qualify the effort the patients submit to as “modeled stress”? Can the authors precise why they chose to submit the study patients to such a strong physical test, when other groups used tests that are milder?

In an effort to address your thoughtful comments above, throughout the paper, we replaced “modeled stress” with a more specific and accurate description of what participants underwent: “exercise challenge.” The specific exercise challenge selected was a standard maximal graded eXercise test based on the McArdle protocol (section 4.1.). This test was utilized largely because the protocol was used as a part of larger ongoing studies of biological mechanisms underlying neuroimmune diseases (added to section 4.1.).

2.There should be a mention in the abstract and the introduction that only female patients were considered, which could induce a gender bias in the results. The authors could add a part on the influence of gender in ME/CFS and the immune system in the discussion to illustrate this issue.

Thank you very much for bringing our attention to this topic. In response, a “limitations” section (section 3.7.) was added to the discussion explaining that our data is limited to females with ME/CFS, and that this is an important limitation, especially because prior studies have shown molecular sex differences between males and females with ME/CFS. In addition, in the introduction (section 1.), we added a few sentences explaining why we focused on female participants with and without ME/CFS. We also restated that the study results only pertain to females. The abstract was also updated to note that the study participants were limited to females, and that our results only apply to females.

Round 2

Reviewer 2 Report

The authors addressed well the comments and issues of the reviewer. However, there are still some concerns.

1)      The introduction should be clearer. Lines 44 to 46, the authors write that some patients are “homebound” when they are subject to an “attack” of ME/CFS. However, if well remembered, one of the issues of the patients is that they are “homebound” in between “attack” periods. This part should be clarified (also not sure about the term “attack”). Moreover, according to the literature, there are questionnaires that help practitioners to diagnose ME/CFS. Hence , writing that “diagnosis relies on the elimination of other conditions” is not totally accurate. Finally, the authors did a great job, justifying the gender of the patients they studied, how the immune system is involved in ME/CFS but this part could be shorten, and the justifications should rather belong to the discussion. The authors could also add one or two sentences about how PBMCs are representative of the immune system and if they lack any of the immune cells evoked in the article.

2)      The authors wrote in the introduction that the “baseline” of the measurements (T0) is located before the exercise challenge. It would be clearer for the reader if T0 was designed as “baseline before exercise challenge” or something similar, in case they did not notice first thing.

3)      Rather than to keep in the results part subsections titles “T2 vs T0”, for example, the authors could use the discussion as a template to have the titles reflect the results found.

4)      It would be interesting to give the cell type abundance table of ME/CFS patients for T1 against T0, at least as supplementary material, just to see that there are no differences.

The miRNA part at the beginning of the discussion is well written but how the authors suppose that miRNA influence gene expression is not clear. Moreover, along the discussion, it is not comprehensible why the authors discussed the pathway changes in healthy individuals and patients independently but still made comparisons between the two groups in these parts. A separation of the parts according to the differences and similarities found between the groups could make more sense.

Author Response

We are very grateful for your comments. We believe that by addressing these comments we made the manuscript more understandable for readers.

Point 1: The introduction should be clearer. Lines 44 to 46, the authors write that some patients are "homebound" when they are subject to an "attack" of ME/CFS. However, if well remembered, one of the issues of the patients is that they are "homebound" in between "attack" periods. This part should be clarified (also not sure about the term "attack"). Moreover, according to the literature, there are questionnaires that help practitioners to diagnose ME/CFS. Hence, writing that "diagnosis relies on the elimination of other conditions" is not totally accurate. Finally, the authors did a great job, justifying the gender of the patients they studied, how the immune system is involved in ME/CFS but this part could be shorten, and the justifications should rather belong to the discussion. The authors could also add one or two sentences about how PBMCs are representative of the immune system and if they lack any of the immune cells evoked in the article.

Response 1: We greatly appreciate your time and insight in providing commentary involving the symptomatology and diagnosis of ME/CFS. As such, in lines 44 to 46, we have reworded this sentence to remove the word “attack”. We have also highlighted how patients experience debilitating malaise during “relapses and remissions of the disease”. In regard to the comment on diagnosis of ME/CFS, we have added a sentence explaining that there are specific criteria that are used to diagnose this disease including “identifying patterns of symptoms”. We believe that these changes clarify the aforementioned statements that may have been misleading.

To address the comment about the justification of gender and the immune system involvement in ME/CFS, we have removed this section from the introduction and added it to section 3.1 in the discussion.

Finally, we have added a sentence regarding what PBMCs are, including what cell types are found in this subset of whole blood cells.

Point 2: The authors wrote in the introduction that the "baseline" of the measurements (TO) is located before the exercise challenge. It would be clearer for the reader if TO was designed as “baseline before exercise challenge" or something similar, in case they did not notice first thing.

Response 2: Thank you very much for your valuable feedback on this matter. In response to this thoughtful commentary, we have rearranged a sentence in the introduction discussing the timepoints we have used in this study to clarify that “baseline before exercise challenge” is T0 and “4-hours after maximal exertion” is T2. Accordingly, we have made this change throughout the rest of the paper.

Point 3: Rather than to keep in the results part subsections titles T2 vs TO", for example, the authors could use the discussion as a template to have the titles reflect the results found.

Response 3: We appreciate your commentary on this, and believe it would be helpful to the reader and other investigators. As such, Section 2.2 of the results has been renamed to “Transcriptomic changes between maximal exertion (T1) and baseline before exercise (T0)”. Section 2.3 of the results has been renamed “Transcriptomic changes between 4-hours after maximal exertion (T2) and maximal exertion (T1)”. This change has also been reflected in the discussion for continuity and so the section titles are reflective of the results found.

Point 4: It would be interesting to give the cell type abundance table of ME/CFS patients for T1 against TO, at least as supplementary material, just to see that there are no differences.

Response 4: Thank you very much for your insightful comment on this section. We agree that this data is a valuable addition to this section of the paper. Therefore, we have added a section to Table 2 labeled “T1 v. T0 in ME/CFS Patients” that reflects the non-significant cell type abundance changes that were seen for ME/CFS patients between maximal exertion v. baseline before exercise.

Point 5: The miRNA part at the beginning of the discussion is well written but how the authors suppose that mRNA influence gene expression is not clear. Moreover, along the discussion, it is not comprehensible why the authors discussed the pathwav changes in healthy individuals and patients independentlv but still made comparisons between the two groups in these parts. A separation of the parts according to the differences and similarities found between the groups could make more sense..

Response 5: We greatly appreciate the insight provided by this commentary. In response, in section 3.4 of the discussion, we have added content from published literature regarding the role of miRNAs in the control of gene expression. We emphasized how these molecules can target and lead to the suppression and, in some cases, destruction of mRNA.

In regards to the comment on how the discussion is structured, we have taken into consideration your comment and eliminated subheadings that separated the time point comparisons by cohort (ME/CFS and HC). Instead, we have combined these sections into a larger segment labeled “Transcriptomic changes between maximal exertion (T1) and baseline before exercise (T0)”  and “Transcriptomic changes between 4-hours after maximal exertion (T2) and maximal exertion (T1)”.  This allows us to have a more logically organized discussion on the similarities and differences between cohorts during each time point comparison.
